# Anxiety, Stress, and Resilience Strategies in Parents of Children with Typical and Late Psychosocial Development: Comparative Analysis

**DOI:** 10.3390/ijerph19042161

**Published:** 2022-02-14

**Authors:** Felicia Andrioni, Claudiu Coman, Roxana-Catalina Ghita, Maria Cristina Bularca, Gabriela Motoi, Ioan-Valentin Fulger

**Affiliations:** 1Faculty of Sciences, University of Petroșani, 332006 Petroșani, Romania; feliciaandrioni@gmail.com (F.A.); vifulger@yahoo.com (I.-V.F.); 2Faculty of Sociology and Communication, Transilvania University of Brasov, 500036 Brașov, Romania; maria-cristina.bularca@student.unitbv.ro; 3Faculty of Social Sciences, University of Craiova, 200585 Craiova, Romania; ac.roxana24@gmail.com (R.-C.G.); gabrielamotoi@yahoo.com (G.M.)

**Keywords:** anxiety, stress, resilience, parents, children, psychosocial development

## Abstract

The child’s developmental characteristics influence the psycho-social features in the behavior of parents. This aspect is relevant in building effective strategies for psychological and socio-educational assistance to parents for an increased quality of family life. The aim of the present study is to investigate the differences in anxiety, stress, and resilience strategies in the case of parents with children with late psychosocial development and those with children with neurotypical development. The research sample consisted of 620 subjects (380 women, 240 men, divided into two equal categories: parents of children with late development and with neurotypical development). The questionnaire survey (Levenstein’s Perceived Stress Questionnaire for stress and Hobfoll’s Strategic Approach to Coping Scale for resilience strategies) and interview (Hamilton’s Anxiety Rating Scale for anxiety) were used. The anxiety levels did not differ significantly (t = 0.45, *p* = 0.65), but there were differences in the perceived stress levels (t = −7.10, *p* = 0.03). As for resilience strategies, significant differences were found for assertive action, social communion, precautionary action, and seeking social support, which were more pronounced strategies that were used by parents of children with late psychosocial development.

## 1. Introduction

This paper is a comparative study with the general objective of recording the differences between parents raising a child with typical psycho-socio-emotional development, who is clinically healthy (both psychologically and physiologically) and the parents of children with late psychological and socio-emotional development, with developmental delays and social adjustment issues. The differences are highlighted on three levels: anxiety, perceived stress, and resilience strategies that are used by parents. Based on the literature and numerous previous studies that have assessed differences in parenting style, parent-child relationship, level of resilience and depression, and parent-child attitudes, this survey aims to investigate the differences in anxiety, perceived stress, and resilience strategies between the two categories of parents within the Romanian population.

This topic, namely parenting style and parent-child relationship, discussed in a general family context, raises many questions nowadays, from the attitude of intrinsic rejection of the parent who has a child with a disability to the daily sacrifice of parents who have children with such conditions. One of the long-term objectives of this research is to emphasize the need for psychosocial intervention on parents that are caring for children with late development in the Romanian health system, which provides specialized services to the child that is affected by a certain disease or disorder, but not to the parents and extended family. The research results can be used in social care and clinical psychology, as well as in social, family, or health psychology. The questions that are raised by the research are: “Are there differences between the level of anxiety and stress perceived by parents of children with typical psychosocial development and parents of children with late psychosocial development?”, “do the parents of children with late psychosocial development choose different coping strategy compared to parents with children with normal psychosocial development?”, “how can one intervene to help these clinically affected parents?”, and “how can they prevent the effects and risks of raising a child with late psychosocial development?”.

The motivation for choosing this topic is the desire to bring a contribution that is applicable in practice in the social intervention field. Although the topic was also studied under various targeted aspects and measured variables, the Romanian population was not studied to the same extent as other populations in the respective specialized studies. The prevalence of autism and developmental disorders has increased dramatically in recent times. This emphasizes the importance of understanding how these conditions have implications and, in turn, affect the parent-child relationship. The interest for the topic is also due to the desire to develop a psychosocial intervention strategy in the form of support groups that are meant to respond to the needs that are revealed by the present study. In other words, the present research is a pillar for further intervention and a much more complex study. Another reason is the desire to improve the life quality of the parents in the target group, an experimental group, at the level of the general population, which further resonates on the parental relationship, respectively, on the harmonious behavioral and emotional development of the child. 

The research objectives of the paper are the analysis of anxiety levels for the parents of children with typical psychosocial development and the parents of children with late psychosocial development, the identification of the perceived stress levels for the two categories, and an analysis of resilience strategies that are adopted by the parents of children with typical psychosocial development compared to the parents raising children with late psychosocial development.

Revealing the impact that the child’s health has on parents and their quality of life, the way they are affected, and how they handle the situation are premises that are absolutely necessary for a good knowledge of the parental profile in crisis situations, enabling an effective intervention in the short or long run.

## 2. Literature Review

### 2.1. Typical and Late Socio-Emotional Development

Children’s social and emotional health is often defined as their ability to develop in the family, community, and cultural context, so that they can form secure relationships, experience and regulate their emotions, and explore and learn. Child development is usually seen as a normal progression through which children change as they grow up by acquiring and refining knowledge, behaviours, and skills. Child development, in general, involves the observance and evaluation of five specific areas: motor/physical development, cognitive, social/emotional development, communication/language, and self-help/adaptation. There are three generally accepted principles of child development that are the rate of development differs among children, development takes place in a relatively orderly process, and this process takes place gradually [1]. As children grow up, various physical and developmental milestones will be reached during each age level, which includes increases in height and weight, as well as development in the five specific areas that are mentioned above.

The terms typical and normal child development are often used interchangeably to refer to children who acquire a wide range of similar skills for most children of the same age, in the same cultural context. However, the term of typical developmental is much largely accepted by parents compared to the normal developmental term. Development not only includes skills that are similar to those of peers, but also involves developmental concepts such as maturation, developmental stages (significant achievement points in various skill areas), skill development sequences, and age-related expectations.

A child with a normal, typical, ongoing psychosocial development acquires specific skills and behaviors based on a predictable rate and sequence. However, no two children develop and grow at exactly the same pace, acquire the same skills at exactly the same time, or exercise those skills in the same way. Therefore, atypical child development or late psychosocial development is a term used to describe children who show marked differences in development to a high degree or whose development appears to be significantly inconsistent with the child’s normal development and/or have significant deviations in normal development sequence. For a better understanding of developmental delay, it is necessary to relate the individual performance of a child to the average performance of a large group of children of the same age. If it performs significantly below average in one or more of the five areas of development, the issue of such a delay may be considered [1].

Emotional development is closely linked to social development, the reformulated text without direct quotations: Emotional development is closely linked to social development, including the expression of a child’s own feelings about others, contextual factors, control of physiological functions, maintaining focused attention, and paying attention in the case of family support and care [2].

### 2.2. Parental Anxiety

Anxiety is presented as an inner state of turmoil that is often accompanied by nervous behaviour, such as rumination and somatic manifestations [3]. Specifically, it is accompanied by muscle tension, concentration problems, the object being not real or not immediately perceivable as fear, and prolonged as a dysfunctional state for a long time and can lead to the development of a generalized anxiety disorder [4,5]. 

The underlying causes of anxiety include a variety of factors, such as family history and family social factors, such as parental rejection, lack of attachment from parental figures, important levels of hostility, authoritarian parenting style, elevated levels of negative affectivity from the mother, modelling of dysfunctional behaviours, drug abuse, and abuse that is present in childhood (emotional, physical, sexual) [6] (p. 530); [7].

As can be seen, anxiety is shaped to a considerable extent in the interactions that man has in his micro-social and macro-social environment, education, and family context being a foundation on which the state of anxiety can be built. Regarding the parent-child relationship in relation to anxiety, it is characterized by a mutual influence: an anxious parent can model a behaviour of such a nature to his child, causing a stable personality trait, but also vice versa.

The parental status, as a social role, may be accompanied by a prominent level of anxiety, which is influenced by a number of other factors, such as the child’s health, the parent’s family background, and the existence of a fearful background in the parent’s experience, all this leading to a more intense state of anxiety and a much higher level, which causes parents to adopt certain patterns of raising their children. Parental anxiety is generally associated with a child’s upbringing style in a hyper-protective manner, which subsequently generates symptoms of anxiety or anxiety disorders in the child, leaving no room for autonomy [8] (p. 618); [9] (p. 542). Hence, parental anxiety is considered to be one of the predictors of childhood anxiety [10].

Another perspective that concerns parental anxiety is caused by the child’s typology as a clinical state of mental health (typical or late psychosocial development). Parents with children with disabilities face a number of obstacles which become a source of anxiety, hyper-protection, rigidity, and cause them to place greater emphasis on family control [11,12] (p. 668). It was found that relatives (parents, grandparents, members of the extended family) from families with multiple incidents of children with autism have high rates of social anxiety [13], and 20% of first-degree relatives of children with autism spectrum disorders they were diagnosed with social anxiety, even before the child was born [14]. Behavioural disorders in children lead to an increase in parental stress, which leads to decreased self-efficacy of parental role and the subsequent development of feelings of depression and anxiety [15] (pp. 8–9).

Socially, they may lose friends and relationships, experience feelings of isolation, anger, sadness, and sometimes even depression. In this case, the parent’s anxiety is primarily related to the child’s vulnerability and the parent’s inability to really help the child. All these correlated elements lead to a state of anxiety, tension, and stress, and the social behavior is one of avoidance and isolation. In other words, the pathology can also be passed down, as the child’s illness can lead to clinically significant changes for the parents or relatives.

At the same time, the emotional state of the adult can affect the quality of life and implicitly of the child’s illness. Parents with elevated levels of anxiety whose children have a chronic disability or condition have been shown to have an inferior quality of life [16,17,18].

### 2.3. Parental Stress

Stress is defined as an atypical biological manifestation of the organism in the face of external factors as well as stressors of various nature, physical, chemical, biological, but also mental [19] (p. 137). This reaction involves an effort of a cognitive and behavioral nature, expressed emotionally, to diminish, control, or tolerate demands that personal resources do not meet [20] (p. 19).

Stress has a negative value and is influenced by negative emotional factors but is also balanced by the individual’s own resources and specific coping or resilience strategies. In relation to the parents and the process of raising, caring for, and educating a child, stress can be generated by many other variables, be it health, school situation, relational or social issues, group integration, and adaptation to the environment. According to statistics, 4% of parents receive stressful news about their child’s health. The birth of a child becomes an ambivalent event for parents, of joy, but also despair and stress [21]. Thus, analyses on parenting stress are very important because they can offer relevant information about the factors which determine the parents’ stress [22].

The parents of a child with a disability or a chronic illness have to deal with many uncertainties about their health, prognosis, repeated medical consultations, and their special care needs [23,24]. Specifically, in the case of children with autism, the emotions that are felt by their parents and the level of stress is much higher and negative, involving hopelessness (60% of fathers and 55% of mothers), anxiety, or strong emotions when the child does not respond to parental love, and parents have to work much harder for them (55% of fathers and 67% of mothers). Other factors intervene also, such as frustration, marital affectation, the feeling that the child always has certain rights violated because he is autistic and unfulfilled expectations about their life as a parent [25,26]. Parents of children with disabilities experience poorer mental health, career disruption, decreased leisure time, and a much higher likelihood of divorce than parents with children without illness or other disabilities [27].

Parents of children with disabilities are subject to higher growth demands compared to parents of children without disabilities, leading to increased levels of stress that is much higher compared to parents of children with neurotypical development [26,28,29,30]. The trajectory of stress in the case of these parents involves an increase in the level of parental stress from early to middle childhood, then a decrease during adolescence. Also, a high level of stress is recorded when the diagnosis is made by the doctor [31] (p. 7).

Cultural differences have been highlighted in terms of the implications for raising a child with disabilities in Western and Asian culture, taking into account how the emotions, cognitions, motivations of people, and parents as a social role differ depending on the cultural environment. Asians value group harmony more, while Western culture values independence [27] (p. 355). When it comes to predictors of parental stress, previous studies have emphasized that the traits that are associated with people with intellectual disabilities, such as medical problems, can be considered elements that contribute to parental stress, even in cases in which other types of family factors are taken into consideration [32]. Culture is also a strong predictor of parental expectations regarding a child’s cognitive and social development; Western cultures expect the child to develop earlier than the developmental charts compared to traditional cultures [33] (p.134).

### 2.4. Parental Resilience

Both mothers and fathers of children with disabilities perceive the prospect of the child’s future as the greatest stressor. It is important for medical and research practice to be based on the recognition of parental individuality in the context of family coping strategies [34] (p. 311). 

Connecting resilience strategies with the differences that have been presented previously between parents with children with typical psychosocial development and parents with children with late psychosocial development, and with disabilities, numerous studies have been conducted that address such differences in coping with stress and side effects through strategies, specifically resilience. Paster, Brandwein, and Walsh [35] (p. 1341) pointed out that there are significant differences between the strategies that are used by parents whose children do not have disabilities, compared to parents who have children with disabilities. The search for social support is a method that is present especially in the case of parents with children with disabilities and the use of more planned problem-solving methods, as well as strategies of avoidance/rejection and positive evaluation. Social support differentiates high-stress mothers from low-stress mothers among those with children with autism, and mothers who perceive social support as more accessible report somatic problems that are associated with stress and much lower depressive symptoms. that number [36]. Moreover, high social support is a resource of resilience for families with children with disabilities, even if they have poor economic status [37].

In addition to the search for social support, the most effective and sanogenic strategies to reduce stress and negative effects in the case of raising a child with disabilities proved to be the strategies that are aimed at solving problems, positive re-evaluations of the given situation, as well as distance in busy moments [38,39,40].

A similar study was conducted on elderly parents, whose average age was 65, who have children with disabilities. The results indicated that strategies for adapting to stress (secondary engagement) that are used especially in later life, reduced the impact of supporting and caring for a person with a disability, while strategies for distracting stress and employment exacerbated the effects of this responsibility on depression. Most of the effects were similar between mothers and fathers, including active stress reduction strategies, which had a greater effect on those with co-resistant children. Vulnerability was highest in those with a co-resilient child who used distraction and disengagement strategies, but those who used engagement and coping strategies were resilient [41].

Another study in this regard was conducted based on the characteristics of family resilience, using as a target group of the parents of children with intellectual, physical, or learning disabilities. It included a wider overview of the issues that were studied. The child’s disability is presented as a triadic experience, involving a three-way interaction among the child experiencing the dysfunction, the family that is affected by it, and the external environment where the dysfunction is manifested [42] (p. 160). The author highlights the pre-existence of a suspicion from the parents (50% of them suspected that their child had a disability due to a condition during birth: ataxia, premature birth, Down syndrome, etc.). Their negative emotional response to the news was associated with feelings of depression, anger, shock, fear, denial, guilt, self-blame, confusion, sadness, despair, hostility, emotional collapse, and later complained of endless fatigue as well as social isolation, the feeling of lack of freedom, and the sentiment that they lack the necessary information related to the education and psychological help necessary for the child. In relation to the extended family, the members’ reactions were divided into three categories: most encouraged the parents, some reacted negatively, expressing indignation at the news, and very few offered practical answers and solutions. As a result, almost half of the parents were worried about the impact that the child with disabilities will have on his or her siblings, who will probably have to take care of such a child; more than a quarter reported financial problems and most stated that their initial negative emotions turned into positive emotions of love and joy. A total of 28% of parents say they still feel anger, frustration, and guilt, especially when comparing their child to others [42] (p. 167).

Also, the resilience mechanisms that are present in the case of parents with children with disabilities are mediated by certain variables. One of them is the age of the respondents/parents. Younger parents (20–29) had a greater tendency to choose spirituality. This may be due to a lack of experience or a desire to control the situation and that is why they accepted the situation and left it in the hands of providence [43] (p. 186). Another variable is the number of children, so that parents with more children had greater financial difficulties. This is due to the fact that with a larger number of children, the responsibilities go to each one and the financial requirements are higher so that the level of stress increases and the ability to cope decreases. Civil status also played a key role, so widows without a life partner adopted the mechanism of resorting to family relationships in dealing with the situation. Married parents did not consider this aspect very important as the presence of the spouse could already help him/her in raising the child [43] (p. 186). Last but not least, the status of ethnic minority can be a factor that differentiates the chosen resilience strategies. Sharing with others, using external support, and recognizing the rewards of caregiving are part of the strategies that are used by the Pakistani minority in the UK, as they are not specific to the population itself, but certain features of the strategies may differ from those that are used by parents with deferred assets [44] (p. 1548).

### 2.5. Additional Characteristics of the Family of the Child with Late Psychosocial Development

The family is generally defined as a major social institution and the environment in which a person spends most of his or her social activities. It is a social unit that is created through blood ties, marriage, and adoption and can be described as being nuclear (parents and children) or extended (including other relatives) [45]. Within the family that has a child with disabilities, this aspect has an impact on all the members, being felt in the family atmosphere, intra-group relationships, and the attitudes of members on an external social level outside the family. Parents, siblings, and extended family all are affected by the fact that a family member has a disability, be it physical, developmental, cognitive etc. 

Reichman, Corman, and Noonan [46] (pp. 679–680) point out some aspects of the family of a child with a late psychosocial development, with a disability, recording several studies. First of all, the existence of a child with a disability or a serious condition in the family increases the risk of the parents divorcing or living separately, the mother generally does not work or relies on social assistance services. Also, the father’s working hours are reduced. The parents of a child with a disability have lower rates of participation in socializing and are less willing to have a large family. 

A family with a child with a disability faces a number of challenges, the stress is amplified by the fatigue that is felt being associated with the low morale of adult family members in relation to the child’s progress, which in the case of neuro-motor disabilities are often lower, compared to parental expectations [42] (p. 167). It is not only the parents who are affected by the upbringing of the child with disabilities in the family. Another category of affected members is that of the siblings, who, in theory, should be on the same level in family relationships as their siblings, but are at an increased risk of not adapting to problems [47]. 

Another valuable and influential feature of the family bringing up a child with disabilities is the socio-economic status, which has proven to be a factor in the resilience of the family of the child with disabilities. Knestirct and Kuchey found in their study that having time to reflect is the key to reconfiguring, positively re-evaluating one’s situation, which is seen as crucial in developing resilience [48]. Once this type of time has been allocated, the family can rebuild its vision of the family, the disability, and the child, but to allocate this time it needs a stable status and free time. Socio-economic status is very relevant, in other words, it has been shown that poverty has an impact on five aspects of the family: health, productivity, physiological environment, emotional well-being, and interaction within the family. Scorgie, Wilgosh, and McDonald’s 25 studies on coping and resilience in families with disabilities have shown that higher monthly income has helped people cope much better [49] (pp. 102–110). 

Moreover, the interaction within the family is also influenced by the socio-economic status. Supportive, caring parenting styles and authoritarian parenting styles have long been associated with positive parenting outcomes and are much more difficult to implement when economic stressors are present [48] (p. 6).

All in all, the data that were collected from previous studies and the literature show significant differences between parents of children with typical psychosocial development compared to parents of children with late development, emotional level, perceived stress, level of depression and anxiety, resilience strategies, and different needs, as well as qualitative characteristics that are present within the family: socio-economic status, marital relations, the experience of siblings growing up with a child with a disability, and the integration of the whole family on the social level. These collected data are the basis of the research hypotheses of this paper and support the need to study the current issue on the Romanian population to establish and develop a specialized intervention plan.

## 3. Materials and Methods

### 3.1. Research Objectives

This paper presents a comparative study, aimed at investigating the differences between the parents of children with late psychosocial development who have a disability, and the parents of children with typical psychosocial development that are clinically healthy, from the perspective of strategies of adaptation to the social reality, on the one hand, and on the other hand from the perspective of the level of anxiety and stress experienced by them.

Based on the results of the analysis and the predictions forwarded, one of the applicable contributions of this research is the development of a model of social and psychological assistance intervention to meet the needs of parents in the case of parents with children with late development and with disabilities, to reduce the perceived stress and anxiety levels and to address healthy resilience strategies.

### 3.2. Research Questions/Hyphotheses of the Research

The present survey starts from the following research questions:Are there significant differences between the anxiety levels of parents of children with late psychosocial development and the parents of children with typical development?Are there significant differences between the level of stress that is perceived by parents of children with late psychosocial development and the parents of children with typical development?Do the resilience strategies that are used by parents of children with typical psychosocial development differ significantly from the resilience strategies that are used by parents of children with late development?

### 3.3. Research Variables

This research identifies an independent variable and three dependent variables in total. They apply separately, depending on the three research hypotheses.

In the case of the first research question there is an independent variable—category nominal variable, the quality of parent of a child with typical/late psychosocial development, and the dependent variable is the measured anxiety level.

In the case of the second research question, the independent variable is also the nominal category variable that is mentioned above, and the dependent variable is the level of perceived stress that is recorded.

As for the third research question, we maintain the same independent variable, the quality of parent of a child with typical/late psychosocial development, and the dependent variable is represented by the type of resilience strategies that are used: assertive action, social communion, search for support, precautionary action, indirect action, instinctive action, aggressive action, and antisocial and avoidance action.

### 3.4. Formation and Selection Criteria of the Sample

The sampling was based on a stratification method that took into account the selection criteria of aspects of social life, family, type of child, age of children, age, and status of parents.

When it comes to the parents that were selected, they had to be married or in a partnership so that they would not be individually involved in raising and educating the child or the level of stress or anxiety would be affected by this issue. 

Secondly, from a clinical point of view, the people that were included in the study had to meet the criteria of not having psychiatric issues in their own medical history, and of not having medication or hospitalizations for psychopathological conditions.

Thirdly, the level of development of the child was taken into account: typical or late, because this aspect represented the independent variable that was involved in the present research, and also the age of the children was taken into account in the case of both categories of parents. Thus, the basic condition was that the parents have children between 2 and 10 years old, so that there is no risk that the level of stress and anxiety will be higher if the parent is with the first child and he is under 2, and the parent has not yet adapted to the requirements and needs of the child or the parental role has not yet been fully mastered. At the same time, the upper limit of 10 years implies an age that requires dependence and involvement on the part of parents in raising and educating him, taking into account the stage of development in which he is.

At the same time, the age of the parents undergoing research between 22 and 55 years was relevant and limited, so that a critical age is not involved, which means specific crises that can affect the level of stress and anxiety investigated from this perspective

### 3.5. Research Population 

The research sample consisted of a total of 620 participants, grouped for testing into 2 categories of subjects, evenly distributed (310 respondents for each category), namely parents with children with late psychosocial development, of which 90 male subjects and 220 female subjects, and parents with children with typical psychosocial development, of which 160 female subjects and 150 male subjects. Depending on the age criterion, which had to be between 22 and 55 years old, the minimum age after data collection was 22 years and the maximum age was 53 years.

The samples were constructed so that they were equal in the number of subjects for greater accuracy: 310 parents of children with typical psychosocial development and 310 parents of children with late psychosocial development.

Another clarification was that the test subjects came from random social backgrounds, regardless of profession, gender, level of education, or other psycho-social issues. Parents of children with late psychosocial development were recruited and contacted through NGOs that provide care and therapy services for such conditions.

### 3.6. Research Methods and Tools

The following research methods were considered to verify the hypotheses: the questionnaire-based survey (Perceived Stress Questionnaire, Levenstein et al., 1993, Stevan E. Hobfoll’s Strategic Coping Approach Scale, 1998); semi-structured interview (Hamilton Anxiety Scale, HRSA); and a comparative analysis using the SPSS statistical program. These methods and tools measured and quantified the dependent variables: anxiety, stress, and resilience strategies that were used.

#### 3.6.1. Hamilton Anxiety Scale 

The Hamilton or HRSA anxiety scale was applied as a 14-item semi-structured interview where the maximum score can be 56. The scale items assess cognitive and somatic symptoms and are given a score of 0 to 4 by the examiner. The points were given as meaning reflect the following meanings: 0—absent, the subject has never experienced these symptoms; 1—weak, the symptoms are manifested irregularly and for short periods of time; 2—moderate, the symptoms are manifested constantly, for certain periods of time and it is necessary the effort of the subject to deal with them; 3—severe, the symptoms are continuous and dominate the life of the subject; and 4—very severe, manifests itself as a disability and hinders the activities of the subject. HARS was designed in 1959 by Max Hamilton and is one of the first tools that was developed to quantify the severity of anxiety symptoms. The scale allows an overall assessment of the psychic (e.g., mental stress, anxious mood) and somatic (e.g., associated bio-physiological changes) symptoms of anxiety. Items that are associated with mental anxiety and those covering somatic aspects of anxiety were established following factor analysis [50] (pp. 51–53). Studies show that people that are diagnosed with generalized anxiety disorder and panic attack have high scores (over 20) on HARS, while people without a clinical diagnosis score significantly lower [51] (pp. 127–134).

#### 3.6.2. Perceived Stress Scale

The Perceived Stress Questionnaire was developed by Levenstein et al. in 1993 and is a questionnaire that establishes the level of perceived stress, completed by self-reporting. The scale contains 30 items, which are rated on a Likert scale from 1 to 4, as follows: 1—almost never; 2—sometimes; 3—often; and 4—almost always. The quotation is made by summing the value given to each item, except for 8 items out of the 30 (1, 7, 10, 13, 17, 21, 25, 29), for which the quotation is reversed. Depending on the score that is obtained, 3 levels of stress can be established: low stress (between 30 and 60), moderate stress (between 60 and 90), and intense stress (between 90 and 120).

#### 3.6.3. Strategic Approach to Coping Scale—SACS

The strategic approach to coping was developed by Stevan E. Hobfoll in 1998 after years of research and is a multiaxial model of resilience, bringing together individual and social aspects of coping [52]. This model includes three dimensions: one prosocial-antisocial, one direct-indirect, and one passive-active [53].

SACS has 52 items and 9 subscales: assertive action, avoidance action, indirect action, seeking social support, social communion, precautionary action, antisocial action, aggressive action, and instinctive action. Each item is rated on a Likert scale from 1 (it is not at all what I would do) to 5 (very similar to what I would do). It is a self-reporting tool, each scale will have a score between 4 and 45. The tool was validated on the Romanian population, on the clinical and non-clinical group with both men and women by Budău et al. [53]. The validation study also confirmed that certain strategies are more often used by people with certain psychological disorders; active and prosocial strategies are generally associated with lower degrees of psychopathology, while avoidance resilience strategies would be associated with high degrees of psychopathology.

### 3.7. Methods of Collecting and Processing Data 

The study participants were verbally instructed on the purpose, hypotheses, and the methods of the research and completed a physical consent form.

The tools were applied in physical, pencil-paper format (stress perception scale and strategic coping approach scale) and face–face interviews were held to collect demographics and complete the Hamilton Anxiety Scale. These took place in an indoor environment, ensuring the confidentiality between the respondent and the interviewer.

The tools were administered by 3 behavioral psychologists from: 2 NGOs in Romania, which offer assistance and therapy services for integration and psychosocial development disorders (Puzzle Association and Senzo Kids Association) and from the individual psychological office. 

After collecting the information, the tools were manually scored by the researchers according to the pre-established calculation guidelines for each questionnaire and for the interview.

The scores and demographics (age, gender, parent category) were entered into the SPSS statistics program (IBM SPSS Statistics Version 26) to test the three research hypotheses.

To test the hypotheses, we used the *t*-test for independent samples, descriptive statistics, and the Levene test for equality of variances, as well as the formula for calculating the effect size-omega-square, which was performed manually.

The t-test for independent samples was used to compare the averages of one variable (in this case anxiety, stress, and resilience) that was measured on two separate samples in terms of another variable (in this case parents of children with typical development and late development) [54] (p. 117).

Descriptive statistics were performed through the descriptive procedure of SPSS and provide information on N (the number of sample values), the minimum and maximum distribution value, the average recorded values, and the standard error (degree of inaccuracy of the two indicators in relation to the actual values in the population from which they were extracted) [54] (p. 278).

The Levene test for equality of variance was calculated in SPSS with the selection of the *t*-test for independent samples. It is necessary to calculate to check if the variance of the values between the two groups is normal. Its result is displayed in SPSS in the table, on the first line. It is necessary to check its value because if the probability of the result is higher than *p* = 0.05, then the variances are equal and the result is read on the first line. Otherwise, equality of variance is not accepted and the result is read on line two [54] (p. 296).

The omega-square calculation formula (Ɯ ^ 2) is used to express the effect size, respectively, of the degree of association between the dependent and the independent variable or how strongly they were associated in the research. Cohen’s interpretation of this value is as follows: 0.01—low association size, 0.06—medium association, and 0.14—high association [54] (p. 297).

## 4. Results

### 4.1. Testing the First Research Question

To test the first research question, the *t*-test was used for independent samples to measure the significance of the difference between the anxiety level of parents with children with typical psychosocial development and the anxiety level of parents with children with late development, recorded after Hamilton anxiety interview (Table 1).

The equality of variances that were tested with the Levene test indicates a probability of 0.17, greater than *p* = 0.05, which means that the variances are equal. Thus, the result indicates t = −0.45, df = 600, *p* = 0.65. Since *p* = 0.65, higher than the threshold of 0.05, the research hypothesis is rejected. In other words, there are no significant differences between the anxiety level of parents with children with typical development and parents with children with late development.

The effect dimension was calculated with the dimension index for the difference between the averages, omega-square, to express the degree of association between the dependent and the independent variable:(1)$$\omega^{2}=\frac{t^{2}-1}{t^{2}+n1+n2-1}=\frac{0.45^{2}-1}{0.45^{2}+310+310-1}=\frac{-0.79}{619.20}=-0.001$$

The −0.001 result indicates a low association dimension, as recommended by Cohen. In other words, there is a low association between the level of anxiety and the type of child that is raised (with typical psychosocial development or with late development). The research data reject the hypothesis 1. 

### 4.2. Testing the Second Question of the Research

To test the second question of the research, the *t*-test was used for independent samples to measure the significance of the difference between the perceived stress level, relative to the child’s developmental level, the level recorded after completing the self-report perceived stress scale (Table 2).

The equality of variances tested with the Levene test indicates a probability of 0.02, less than *p* = 0.05, which means that the variances are not equal. Thus, reading on the second line, the result indicates t = −7.10, df = 605, *p* = 0.031. Since *p* = 0.031, lower than the threshold of 0.05, the research hypothesis is confirmed. In other words, there are significant differences between the perceived stress level of parents with children with typical psychosocial development and parents with children with late psychosocial development.

The effect dimension was calculated with the dimension index for the difference between the averages, omega-square, to express the degree of association between the dependent and the independent variable:(2)$$\omega^{2}=\frac{t^{2}-1}{t^{2}+n1+n2-1}=\frac{-7.10^{2}-1}{-7.10^{2}+310+310-1}=\frac{-49.41}{-50.41+619}=0.07$$

The result of 0.07 indicates an average dimension of the association, according to Cohen’s recommendation. In other words, there is a moderate association between the perceived stress level and the type of child that is raised (with typical or late psychosocial development). 

### 4.3. Testing the Third Research Question

To test the third questions of the research, the *t*-test was used for independent samples to measure the significance of the difference between the resilience strategies that were used by parents of children with late psychosocial development and the strategies that were used by parents of children with typical development SACS (Table 3).

The *t*-test was applied to the independent samples for each subscale that was targeted by the instrument that was used, SACS: assertive action, seeking social support, precautionary action, social communion, indirect action, antisocial action, avoidance, aggressive action, and instinctive action. Thus, the data were analysed statistically for each targeted subscale.

For the assertive action subscale the equality of variances that were tested with the Levene Test indicates a probability of 0.02, less than *p* = 0.05, which means that the variances are not equal. Thus, reading on the second line, the result indicates t = −2.95, df = 600.78, *p* = 0.031. Since *p* = 0.031, less than the threshold of 0.05, the research question is confirmed. In other words, there are significant differences between the degree of assertive action as a resilience strategy on the part of parents of children with typical psychosocial development and the parents of children with late psychosocial development.

For the social communion subscale, the equality of variances that were tested with the Levene test indicates a probability of 0.33, higher than *p* = 0.05, which means that the variances are equal. Thus, reading on the first line, the result indicates t = −4.94, df = 618, *p* = 0.000. Since *p* = 0.000, less than the threshold of 0.05, the research question is confirmed. In other words, there are significant differences between the degree of social communion as a resilience strategy from the parents of children with typical psychosocial development and the parents of children with late psychosocial development.

For the precautionary action subscale, the equality of variances that were tested with the Levene test indicates a probability of 0.036, less than *p* = 0.05, which means that the variances are not equal. Thus, reading the second line, the result indicates t = −6.86, df = 618, *p* = 0.000. Since *p* = 0.000, less than the threshold of 0.05, the research question for the precautionary action subscale is confirmed. In other words, there are significant differences between the degree of precautionary action as a resilience strategy on the part of parents of children with typical psychosocial development and the parents of children with late psychosocial development.

For the aggressive action subscale, the equality of variances that were tested with the Levene test indicates a probability of 0.036, less than *p* = 0.05, which means that the variances are not equal. Thus, reading on the second line, the result indicates t = −0.29, df = 616.04, *p* = 0.77. Since *p* = 0.77, higher than the threshold of 0.05, the research question for the aggressive action subscale is refuted. In other words, there are no significant differences between the degree of aggressive action as a social resilience strategy on the part of parents of children with typical psychosocial development and the parents of children with late psychosocial development.

For the avoidance subscale, the equality of variances that were tested with the Levene test indicates a probability of 0.00, less than *p* = 0.05, which means that the variances are not equal. Thus, reading on the second line, the result indicates t = 2.45, df = 540.31, *p* = 0.14. Since *p* = 0.14, higher than the threshold of 0.05, the research question for the avoidance subscale is refuted. In other words, there are no significant differences between the degree of avoidance as a resilience strategy from the parents of children with normal psychosocial development and the parents of children with late development.

For the indirect action subscale, the equality of variances that were tested with the Levene test indicates a probability of 0.00, less than *p* = 0.05, which means that the variances are not equal. Thus, reading on the second line, the result indicates t = 1.24, df = 552.82, *p* = 0.21. Since *p* = 0.21, higher than the threshold of 0.05, the research question for the indirect action subscale is refuted. In other words, there are no significant differences between the degree of indirect action as a resilience strategy on the part of parents of children with typical psychosocial development and the parents of children with late psychosocial development.

For the instinctive action subscale, the equality of variances that were tested with the Levene test indicates a probability of 0.94, higher than *p* = 0.05, which means that the variances are equal. Thus, reading on the first line, the result indicates t = −2.72, df = 618, *p* = 0.07. Since *p* = 0.07, higher than the threshold of 0.05, the research question for the instinctive action subscale is refuted. In other words, there are no significant differences between the degree of instinctive action as a resilience strategy on the part of parents of children with typical psychosocial development and the parents of children with late psychosocial development.

For the social support search subscale, the equality of variances that were tested with the Levene test indicates a probability of 0.18, higher than *p* = 0.05, which means that the variances are equal. Thus, reading on the first line, the result indicates t = −5.22, df = 618, *p* = 0.00. Since *p* = 0.00, less than the threshold of 0.05, the research question is confirmed. In other words, there are significant differences between the degree of seeking social support as a resilience strategy from the parents of children with typical psychosocial development and the parents of children with late psychosocial development.

For the antisocial action subscale, the equality of variances that were tested with the Levene test indicates a probability of 0.01, less than *p* = 0.05, which means that the variances are not equal. Thus, reading on the second line, the result indicates t = 3.74, df = 602.67, *p* = 0.77. Since *p* = 0.77, higher than the threshold of 0.05, the research question for the antisocial action subscale is refuted. In other words, there are no significant differences between the degree of antisocial action as a resilience strategy from the parents of children with typical psychosocial development and the parents of children with late development.

For the subscales for which the research questions/hypotheses of the research were confirmed, with significant differences, the averages of the scores that were obtained from the descriptive statistics tables were analyzed (Table 4).

Thus, for the assertive action, the parents of children with late development registered a higher average (m = 35.32) compared to the parents of children with typical development (m = 33.84). For social communion, the same difference in scores was observed where the parents of children with late development obtained a higher score (m = 17.74) compared to the parents of children with typical development (m = 16.13). For cautious action, the average of parents with children with late development was higher (m = 17.84), compared to those with typical development (m = 15.94). The same report is valid for the search for social support (for parents with children with a late development m = 25.13 and those with children with typical development have an average of m = 22.68).

## 5. Conclusions

As shown by the statistical analysis of the data subjected to the present research, the first hypothesis of the research is refuted, so there are no significant differences between the level of anxiety that was reported by the parents of children with typical psychosocial development and the parents of children with late psychosocial development.

In other words, anxiety as a condition in the parenting process manifests itself at similar levels, regardless of the type of child that is raised, even if the child is clinically healthy or has a disability or other developmental disorders. An important aspect of this research is the fact that the age of the children whose parents were questioned was between 2 and 10 years so a child age was characterized by dependence on parents, which may correlate with the levels of concern because at an early age, the child, whether healthy or not, requires a high degree of involvement, protection, and attention. Furthermore, considering the aspects that are mentioned above and the results of a previous study which revealed that the mothers of children without developmental disabilities registered higher levels of stress than mothers of children with developmental disabilities [55], we argue that when conducting such studies more attention should be paid to the general psychological conditions of parents. 

Previous studies that were conducted on parents, especially on mothers with children with disabilities, disabilities which can be a source of anxiety for parents [11] revealed by high levels of parenting anxiety in mothers. Such levels of anxiety can affect the parenting behavior, parents registering low levels of warmth, and positive affection towards children [56]. Anxiety as a short-term condition can become a stable trait over time and it can be shaped to a considerable extent in the interactions that man has in his social environment, micro and macro, education, and family, favoring the emergence and stabilization of this trait.

A contextual situation that is related to the stage of psychosocial development of the individual is the status of parent, which can involve anxiety, both in the form of a state and in the form of a stable long-term trait. Of course, this is influenced by a number of other factors, such as the child’s health, the parent’s family background, and the existence of a fearful background in the parent’s experience, all of which lead to a more intense state of anxiety and to a much higher level, which causes parents to adopt certain patterns when raising their children.

As Clarke, Cooper, and Creswell [8] have argued, parental anxiety is generally associated with a child-rearing style in a hyper-protective manner, but given the age of the children, parental fears are similar and common for those who have to raise a clinically healthy or an unhealthy child, both require care and dependence. Among the common fears are: the fear of not hurting the child, the very high attention accompanied by hypervigilance towards food, medication and hygiene, the need to be in the visual circle of the parent if he goes out to play, and if he is left alone in bed when small or left alone in the room etc.

Therefore, the fears and the level of self-reported anxiety of the parents are similar, regardless of the characteristics of the child that is raised, whether he is clinically healthy or not, the early stage of development in which the child is and the level of dependence that the process parenting requires a similar degree of parental anxiety.

With regards to the second hypothesis, the level of stress that is perceived by parents with children with typical psychosocial development and parents with children with late psychosocial development differs significantly. In other words, the child’s characteristics in terms of health and illness are predictors of the level of perceived stress from the parents, emphasizing that the child’s health involves a higher percentage of stress on the parent, as suggested and previous studies [57,58,59].

The differences between the levels of stress that is recorded by the parents of children with developmental difficulties and the parents of children with typical psychosocial development may be due to the demands for upbringing and care, which are much higher in the case of affected children [28]. Such demands may refer to higher financial and social demands, time resources, and the level of stress may be intuitively more pronounced for parents of children with disabilities. 

On the social level, parents may lose friendships, they may have feelings of isolation, anger, sadness, and sometimes even depression. In this case, the parent’s anxiety is primarily related to the child’s vulnerability and the parent’s inability to really help him. All these correlated elements lead to a state of tension and stress. Social isolation, which can be a major stressor, is caused by issues such as the parent’s difficulty of traveling with the child, attending events involving the child’s interaction with the elderly, even prolonged hospitalizations due to existing co-morbid conditions [60]. Moreover, some families feel social isolation as a result of stigmatization of the child and of them as parents [61].

Finding a community that encourages, supports, and includes the family of a child with special needs is another element that leads to significant isolation and stress. Many parents are reluctant to attend community events because they feel that the feedback from others is not worth the effort to try to attend events. However, those who find an inclusive community receive meaningful emotional support that helps them reinterpret their family and their role as parents. 

In the case of intellectual or psychosocial developmental disorders such as autism, there is a certain superficial stigmatization as the disability is not visible to the naked eye and the child’s manifestations in public can be misunderstood by people. Children are seen as naughty or disobedient, and their parents are seen as people who are unable to control them or educate them properly. This leads to the isolation not only of the child with autism, but also of the siblings and of the other members of the family who are barricading themselves, such behaviour having an impact on their emotional and social well-being. Hence, caring for a child with autism or another disability requires a considerable amount of financial, emotional, and physical effort. In general, parents feel guilty for isolating their child, at the same time for being judged by society. In many cases one of the parents does not work to be able to take proper care of the child and often these aspects can even lead to the divorce of the parents. Even more so, the continuous supervision involves a great deal of physical effort as many people with autism have difficulty sleeping and eating. As children grow up it becomes more and more difficult to control them when they have a tantrum, and thus parents become discouraged, helpless, and more prone to giving up [62].

Furthermore, the moment when the child is diagnosed represents a major stress factor for the parent. Once the diagnosis is found, subsequent thoughts, strategies, and worries appear, regarding the financial aspects, the child’s future, his social inclusion, the opinion of those around him, from the close family to the extended one, and the members of the society. All these aspects that occur with the diagnosis of the child determine a level of major stress on the parent. Also, financial difficulties as the child’s condition requires a financial effort, will determine the occurrence of stress in other aspects of life and the quality of life in general may be affected later.

When it comes to the third hypothesis of the research, some of the resilience strategies were different in terms of the two categories of parents. As has been observed, parents of children with late development are more likely to resort to strategies to manage stress and difficulties such as assertive action, social communion, precautionary action, and social support search.

Taking into account Hobfoll’s approach to precautionary action as a resilience strategy, [52] it is a pro-social, active strategy that involves a set of environmentally conscious behaviours before acting. Self-sufficient models see this cautious strategy as a sign of weakness, while Eastern European cultural models see this strategy of analyzing the environment before acting as very effective. In the case of this research, the precautionary action can be interpreted in both directions: it can be a sign of weakness where the parent who has a child with late development will be sensitive to every detail of the environment to protect it, will feel obliged to behave cautiously to understand and accept the child or to be hyper-vigilant to prevent any aspect that may affect the child, hinder the parenting process in a positive sense of caution and acceptance of the situation.

Precautionary action is the strategy that is aimed at preventing environmental variables in the active sense, so as to mitigate risk situations or negative events. Parents bringing up a child with a disability will be much more attentive and analytical to aspects involved in their lives and those of their child, whether it is food, hygiene, medication, and the social, school environment of the child’s inclusion. The child’s characteristics of illness and health will be predictors of the parent’s way of acting, in the sense of active pro-social behaviour, which can also be a defense mechanism against helplessness and a feeling of inferiority to others. Therefore, the active prosocial behaviour of the parents of children with late psychosocial development is an adaptive strategy that is useful in the face of the parental process, but at the same time, it can hide feelings of inferiority and helplessness.

At the same time, assertiveness, the need for social communion, and the search for social support are strategies that have proven to be much more accented among parents with children with disabilities or a developmental impairment [62] (p. 42). Social support plays a buffer role in the face of parental stress, preventing possible clinical conditions such as depression or anxiety. Different authors believe that people are born and coexist in social networks such as family, school, interest groups, professional groups, etc., which is also an environment of support in sensitive moments or crises in the personal life [63,64,65]. It is about both the emotional support that is provided and the practical means that human relationships offer to the individual to solve some of the life problems that they face. In the case of parents with children with late development, there may be implications such as recommendations from doctors, therapists, exchange of ideas and good practices, the need to be understood, and to receive compassion from others. The opposite of looking for community and support from others can mean isolation, helplessness, depression, and maybe even suicide. The group offers the opportunity for change, motivation, and encouragement in the face of demanding situations, helping to build personal resilience.

Considering that sociology is a science that studies social human life and social processes, this study may be relevant and applicable in this field too. A prolonged psychological intervention on vulnerable groups of parents of children with late psychosocial development could lead to an increase in the quality of life of these individuals, undoubtedly influencing the social process of raising and caring for the child, integration, and social adaptation of people in this specific category. 

The present study, although it investigates psychological variables such as stress, anxiety, and resilience, has implications regarding human behaviour in social relationships and regarding the dynamic relationships of these vulnerable groups (atypical children and their relatives) with the community that they belong to. Thus, there is a fine line between the sociological and psychological approach, with the two being related in understanding the family issues that the pre-existence of a child’s disease implies at the individual and social level.

Hence, this research falls within the field of clinical psychology as it measures clinical aspects of family life: anxiety, stress, and strategies for coping with stress, but it can also be included in the category of social psychology because it also focuses on aspects regarding social difference, or labelling, and on the way these aspects may affect the individual.

In conclusion, parents who are actively involved in the parenting process of a child that is aged 2–10 years, an early age, which requires the child’s involvement and dependence on the parent, will be equally fearful and anxious and will present similar fears, regardless of the child’s characteristics of illness and health. However, the child’s characteristics will cause long-term worries, miscellaneous expenses, and various financial issues, which will affect the level of stress and the quality of life of the parent involved in the parenting process. Last but not least, in the parental process, stress and daily events will be confronted by parents with children with late psychosocial development through prosocial means of assertiveness, social support search, and communion, will act more cautiously and will show active behaviour as a way of compensating for the situation of raising the child that requires extra effort.

### 5.1. Limitations of the Research

At the basis of the statistical results, it is possible that there were a number of factors and aspects that influenced them. Some of these can be deduced directly while other aspects may be relative or possible error factors. 

Of these, the following (possible) limitations can be mentioned: 

Gender differentiation. The results would have been more concrete if sampling or a gender-based comparison had been made by studying the variables separately on mothers and fathers. Gender is a variable that also implies social differences in the child’s upbringing, which is why it could be an aspect to remember in the future.

The truth of the answers may be that the parents undergoing the test may not have been completely honest in the one-on-one interview, especially with regard to intimate matters.

Controlling the variable that is related to the parents of children with disabilities, in the sense of establishing the exact type of disability, as each disease condition from the child may come with different requirements and demands socially, financially, and physically. The child’s disability can act as a mediator for the level of anxiety and stress or the resilience strategy that is chosen by the parents.

The weaknesses that are listed are aspects to be considered in the future to better control both the replication of the study and its continuation or to enrich the statistical and effective complexity at the scientific level.

### 5.2. Openings to Other Directions of Research

The present study can be classified in the field of sociology as it aims at the quality of family life and clinical psychology, and it measures the clinical aspects of family life: anxiety, stress, and the strategies to deal with stress. However, it may also be included in the category of social psychology, targeting the aspects that social difference, labelling can cause at the level of the individual.

Considering that sociology is a science that studies social human life and social processes, this study may be relevant and applicable in this field too. A prolonged psychological intervention on vulnerable groups of parents of children with late psychosocial development could lead to an increase in the quality of life of these individuals, undoubtedly influencing the social process of raising and caring for the child, integration, and social adaptation of people in this specific category. The main aspect for the future would be the elaboration of a therapeutic intervention plan which would smooth out and optimize the registered negative aspects. This can also be done in the form of a test and a re-test, starting from the conclusions of the present study, on the support groups for parents of children with late psychosocial development and the registration of variables such as stress and anxiety. One can also set clear goals and optimize the quality of life by learning relaxation techniques, with better personal time management, so that there is free time, new ways of expressing emotions, and assertive communication in couples etc.

For further research, it would be important to understand in which way the stress or anxiety levels in parents of children with no atypical development is, without doubt, due to this condition and no to any other problems. For a better clarification of this causal relationship, an investigation in the form of a sociological survey regarding various aspects of quality of life: financial status, occupation, living conditions, extended family relationships, and social support would be needed. All these elements can, in turn, be variables that can influence, and even accentuate, the level of stress and anxiety in the case of parents who already have a condition that makes it difficult for them to integrate and function socially and emotionally with them having children with atypical development. An interesting and very important aspect to be tested in relation to the studied issue would be the evaluation of the attachment style and the type of relationship that both mother and father have with the child, as the key variable is the child’s health characteristic. One can also add here the direct observation of the parent and the child in a natural environment such as the park or a playground to observe the parent-child interaction and the attitude of the parent in direct connection with the child and the social environment. 

Furthermore, when studying the level of stress and anxiety of parents of children with and without disabilities, future research should examine more thoroughly the general psychological conditions of the parents.

Of course, as mentioned before, another aspect of openness would be a model of mediation, the observation of the role of variables and whether one is the mediator of the other two and the addition of new variables in research, such as quality of life, depression level, attachment style, negative automatic thoughts, and cognitive patterns, etc. All this can enrich the profile of such a parent and can develop the direction of therapeutic intervention.

To conclude, there are many directions for expanding, opening and researching this topic, and the present conclusions are a foundation for a first primary therapeutic intervention.

## Figures and Tables

**Table 1 ijerph-19-02161-t001:** *t*-test for independent sampling for the anxiety level.

Independent Samples Test										
		Levene’s Test for Equality of Variances		*t*-Test for Equality of Means						
		F	Sig.	t	df	Sig. (2-Tailed)	Mean Difference	Std. Error Difference	95% Confidence Interval of the Difference	
									Lower	Upper
Anxiety	Equal variances assumed	1.859	0.178	−0.455	600	0.651	−1.161	2.552	−6.267	3.944
	Equal variances not assumed			−0.455	586.46	0.651	−1.161	2.552	−6.267	3.944

**Table 2 ijerph-19-02161-t002:** *t*-test for independent samples for the perceived stress level.

Independent Samples Test										
		Levene’s Test for Equality of Variances		*t*-Test for Equality of Means						
		F	Sig.	t	df	Sig. (2-Tailed)	Mean Difference	Std. Error Difference	95% Confidence Interval of the Difference	
									Lower	Upper
Perceived stress	Equal variances assumed	5.117	0.024	−7.101	618	0.031	−9.968	1.404	−12.724	−7.211
	Equal variances not assumed			−7.101	605.731	0.031	−9.968	1.404	−12.724	−7.211

**Table 3 ijerph-19-02161-t003:** *t*-test for independent samples for resilience strategies.

Independent Samples Test										
		Levene’s Test for Equality of Variances		*t*-Test for Equality of Means						
		F	Sig.	t	df	Sig. (2-Tailed)	Mean Difference	Std. Error Difference	95% Confidence Interval of the Difference	
									Lower	Upper
Assertive action	Equal variances assumed	4.998	0.026	−2.953	618	0.003	−1.484	0.503	−2.471	−0.497
	Equal variances not assumed			−2.953	600.789	0.003	−1.484	0.503	−2.471	−0.497
Social communion	Equal variances assumed	0.927	0.336	−4.949	618	0.000	−1.613	0.326	−2.253	−0.973
	Equal variances not assumed			−4.949	614.787	0.000	−1.613	0.326	−2.253	−0.973
Precautionary action	Equal variances assumed	4.411	0.036	−6.866	618	0.000	−1.903	0.277	−2.448	−1.359
	Equal variances not assumed			−6.866	618.000	0.000	−1.903	0.277	−2.448	−1.359
Aggressive action	Equal variances assumed	0.838	0.360	−0.292	618	0.771	−0.0097	0.332	−0.748	0.555
	Equal variances not assumed			−0.292	616.047	0.771	−0.0097	0.332	−0.748	0.555
Avoidance	Equal variances assumed	59.318	0.000	2.456	618	0.014	0.0968	0.394	0.194	1.741
	Equal variances not assumed			2.456	540.314	0.014	0.968	0.394	0.194	1.742
Indirect action	Equal variances assumed	43.150	0.000	1.244	618	0.214	0.323	0.259	−0.187	0.832
	Equal variances not assumed			1.244	552.826	0.214	0.323	0.259	−0.187	0.832
Instinctive action	Equal variances assumed	0.004	0.948	−2.727	618	0.007	-.0935	0.343	−1.609	−0.262
	Equal variances not assumed			−2.727	615.298	0.007	−0.0935	0.343	−1.609	−0.262
Support search	Equal variances assumed	1.753	0.186	−5.228	618	0.000	−2.452	0.469	−3.372	−1.531
	Equal variances not assumed			−5.228	617.343	0.000	−2.452	0.469	−3.372	−1.531
Antisocial action	Equal variances assumed	5.672	0.018	3.747	618	0.770	1.387	0.370	0.660	2.114
	Equal variances not assumed			3.747	602.673	0.770	1.387	0.370	0.660	2.114

**Table 4 ijerph-19-02161-t004:** Descriptive statistics (resilience scales whose hypothesis has been confirmed).

Group Statistics					
	Type of Parent	N	Mean	Std. Deviation	Std. Error Mean
Assertive action	Parent_child_typical	310	33.84	6.765	0.384
	Parent_child_atypical	310	35.32	5.702	0.324
Social communion	Parent_child_typical	310	16.13	4.202	0.239
	Parent_child_atypical	310	17.74	3.908	0.222
Precautionary action	Parent_child_typical	310	15.94	3.450	0.196
	Parent_child_atypical	310	17.84	3.452	0.196
Support search	Parent_child_typical	310	22.68	5.742	0.326
	Parent_child_atypical	310	25.13	5.932	0.337

## Data Availability

No new data were created or analyzed in this study. Data sharing is not applicable to this article.

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
