# Peer review of "Anxiety, Stress, and Resilience Strategies in Parents of Children with Typical and Late Psychosocial Development: Comparative Analysis"

_ijerph, 2022, doi:10.3390/ijerph19042161_

Round 1

Reviewer 1 Report

Dear Sir/Mam

Please find bellow the requested review regarding the manuscript. The article contains a lot of useful information on the issue. The topic is very interesting and use of sources is appropriate. In addition, tables are very useful although some of them contain too much information.

The article contains a lot of useful information on the issue. It is quite clear what is already known about this topic and the research question is clearly outlined. The abstract is too brief and Discussion section involve too much information. There must be a balance in the manuscript

Specifically

Literature review

Literature review section is too long, while Discussion section doesn’t involve too much information. There is an asymmetry in the manuscript.

Research methods and tools

Who administered the tools?

Results

Instead of t-test try using one way ANOVA

Positive: There are some strengths of the article that could have an impact in the field, such as the topic and its impact on the existed literature. The manuscript is approved after major changes.

Reviewer 2 Report

It would be important to understand in which way the stress or anxiety levels in parents of children with no typical development is without doubts due to this condition and no to any other problems

About the choose of sample there are poor information regarding the general "clinical" status of people involved (if they have had for ezample anxiety problems in tha past)

line 621-622 . this is te reason way general psychological  conditions of parents must be deeply investigated

Important to specify the difference between a sociological approach and a psychological one, as referred in the conclusion which are not clearly connected with the entire research

Reviewer 3 Report

Dear authors

thank you for this research. Overall the paper is well written, the procedures are well described. I suggest reading and minor spell check; avoid directs quotations  at the beginning at a paragraph or sentence and , if possible, reduce the number o direct quotations. Please remove the bullets in the limitations; some references need to be updated. 

Round 2

Reviewer 1 Report

I agree